# Stability of Supehydrophobic Layers Formed by Organic Acids on the Surface of Aluminum Alloy 6063

Alexey M. Semiletov [ID], Alexander A. Chirkunov *, Oleg Yu. Grafov [ID] and Yurii I. Kuznetsov

Laboratory of the Physicochemical Principles of Inhibition of Metal Corrosion, A.N. Frumkin Institute of Physical Chemistry and Electrochemistry, Leninsky Prospect 31 bldg. 4, 119071 Moscow, Russia
* Correspondence: chirkunov@inbox.ru

**Abstract:** The paper discusses the possibility of obtaining a uniformly inhomogeneous surface of aluminum alloy 6063 as a result of alkaline etching and laser processing. Further surface treatment with ethanol solutions of octadecylphosphonic (ODPA) and stearic acids leads to its superhydrophobization (SHP). The study of the degradation kinetics of SHP coatings in water and under conditions of neutral salt spray showed the high stability of ODPA films obtained on a laser-textured surface with an irregularities height of 9.82 μm. X-ray photoelectron spectroscopy (XPS) results showed that ODPA is chemisorbed on the alloy surface. High corrosion resistance of the surface with superhydrophobic layers confirmed by polarization measurements, electrochemical impedance spectroscopy (EIS) and corrosion tests.

**Keywords:** corrosion; aluminum alloy; superhydrophobicity; organic acid





## 1. Introduction

Aluminum alloys are the most important structural materials that are widely used in industry. Due to the high affinity of aluminum for oxygen, its alloys are always covered with a thin protective oxide film, which determines their relatively high corrosion resistance. However, in a humid atmosphere, the resistance of aluminum alloys decreases, and in many cases, local corrosion occurs, for example, under the influence of chlorides [1].

To reduce the corrosion rate of aluminum alloys in various environments, anodizing [2], conversion [3] and plasma electrolytic oxidation (PEO) coatings [4,5], sol-gel [6] and layered double hydroxides technologies [7] are used as effective means of protection. One of the most common, simple and affordable methods of protection is the use of corrosion inhibitors (CIs). Chromium (VI) compounds were previously used as effective CI of aluminum and its alloys; however, due to the tightening of environmental requirements, the use of chromate-containing CIs is limited [8], and they are replaced by inorganic CIs, such as permanganates and molybdates [9,10], and by organic CIs [11,12].

The ability of organic molecules to self-organize on the surface of aluminum alloys can lead to their hydrophobization, which contributes to an increase in corrosion resistance [13,14]. Recently, superhydrophobic (SHP) coatings on the surface of various metals and alloys have attracted much attention of researchers. Due to the various application possibilities, SHP coatings can be used to obtain self-cleaning and more corrosion-resistant surfaces [15–19], as well as the surfaces with anti-biofouling [20] and anti-icing properties [21].

The surface of many metals, including aluminum, is hydrophilic; therefore, in order to give them hydrophobic properties, it is necessary to take into account the surface morphology. Thus, in order to achieve the SHP state, at the first stage a surface with a multimodal roughness is obtained, which is subsequently modified with compounds with low surface energy having alkyl and perfluorinated chains [22,23]. An available method for obtaining multimodal roughness is chemical etching in solutions of acids [24,25] and alkalis [26,27]. However, despite the simplicity and low cost of etching, it is rather difficult to control the formation of ordered structures on the metal surface.

Treatment of the metal surface with microstructure roughness in solutions of hydrophobic agents leads to the formation of air layer that significantly reduces the contact area between the drop and the metal surface. Thus, SHP surfaces can inhibit atmospheric corrosion by preventing the formation of an electrolyte film [15–19]. It should be noted that the water-repellent properties of SHP surfaces depend on the relative sizes of moisture droplets and surface microstructures [28].

With a slight slope of the surface, water droplets can slide off due to the non-wettability of air trapped by irregularities [22,29]. However, this self-cleaning effect of SHP surfaces depends on the size of the microstructures. Drops of saline solution can roll off the surfaces with small microstructures ($\approx$5 μm); at the same time, they can penetrate deep into large microstructures ($\approx$30 μm), which leads to a decrease in corrosion resistance [29,30]. Moreover, microdroplets deposited from the real marine atmosphere can be very small and directly condense in the microstructures of the surface, as a result of which an air layer may not be formed. As soon as the aqueous solution penetrates the air barrier after prolonged immersion, the Cassie wetting model transforms into the Wenzel wetting model, in which the contact between the droplet and the surface becomes much larger than the droplet size, and thus, the anti-corrosive properties of the SHP surface are reduced [15].

In practice, laser ablation is an effective method for the formation of ordered microstructures on a metal surface [31–33]. The shape and size of the structures obtained by this method can be varied by adjusting the parameters of laser processing: laser power, frequency, pulse duration and processing speed. For example, it was shown in [33] that after laser treatment on the surface of the AMg3 aluminum alloy, periodic structures of aluminum oxide are formed. Subsequent modification of this surface in a decane solution of fluorosilane forms an SHP coating with a high contact angle ($\theta_c$) 173.1°–173.4° and a small roll-off angle 2.1°–2.4°. In addition, the obtained SHP coating effectively inhibits the corrosion of the alloy in chloride solutions.

Non-toxic higher carboxylic acids [34], for example, stearic acid (SA) [35], or phosphonic acids [36–38] can be an alternative to the use of expensive fluorine-containing compounds to achieve the SHP state. Since alkyl phosphonic acids R-P(O)(OH)$_2$ are dibasic, they are able to form chelate complexes with cations of different metals. It was shown in [39,40] that alkyl phosphonic acids are spontaneously adsorbed on the oxidized metal surface through the formation of tridentate complexes with surface hydroxides. The authors of [41] reported on the possibility of obtaining SHP coatings on the anodized surface of aluminum alloys with subsequent treatment in aqueous-ethanol solutions of tetradecanephosphonic acid.

Octadecylphosphonic acid (ODPA) was successfully used to obtain self-organizing monolayers on an oxidized metal surface [42–44]. Its effectiveness is due to the fact that it can form strong covalent bonds with metal oxides, as well as compact adsorption films due to the van der Waals interaction between the long alkyl chains of its molecules. In addition, having a molecule length of $\approx$2.2 nm, ODPA can be packed into self-organizing bilayers [45].

Since the production of SHP surfaces is actively gaining popularity, including anti-corrosion protection, it is of interest to study their stability under the influence of corrosive factors.

Maintaining the water-repellent properties of the films under such conditions limits the contact of the surface with a corrosive environment (condensed water, salt solutions) and prolongs the protection period. However, the loss of the superhydrophobic state does not necessarily mean immediate corrosion, since a protective film on the surface may remain, although not well organized to retain the hydrophobic properties.

Aluminum alloys are widely used in various industries and are often exposed to corrosive factors. In particular, alloy 6063 finds application in architectural and building products, in electrical components and conduits, in pipes and tubes and in door and window frames. The use of superhydrophobic coatings on aluminum alloys makes it possible to give them not only anti-corrosion properties, but also the ability to self-clean, and also prevents fouling by microorganisms.

In this work, we continue studies on the modification of the surface of alloy 6063 with solutions of ODPA and SA for its protection against atmospheric corrosion, which were started earlier [46]. The purpose of the article is to evaluate the anti-corrosive properties and stability of SHP layers obtained on alloy 6063 after alkaline etching and laser treatment.

## 2. Experimental

### 2.1. Materials

The studies were performed using samples of aluminum alloy 6063 (UC Rusal, Moscow, Russia), with the following chemical composition in weight, %: Fe $\leq$ 0.35, Si $-$ 0.2 $\div$ 0.6, Mn $\leq$ 0.1, Cr $\leq$ 0.1, Ti $\leq$ 0.10, Cu $\leq$ 0.1, Mg 0.45 $\div$ 0.9, Zn $\leq$ 0.1, with the rest being Al. The samples were the plates with dimensions $30 \times 40 \times 2$ mm$^3$. The sheets of alloy were mechanically polished with sandpaper (400, 600, 800, 1000, 1500 grit) and then degreased with ethanol and cleaned in distilled water and dried in air at room temperature ($t = 20 \pm 2$ °C).

### 2.2. The Preparation of SHP Surface

To obtain a uniformly inhomogeneous rough surface, specimens of alloy 6063 were subjected to:

- Short-term etching (10 s) in a 10% NaOH solution (analytical grade, RusKhim, Russia) at $t = 65$ °C, followed by washing the samples with distilled water and drying in air at $t = 65$ °C.
- Processing with an air-cooled fiber-optic laser marker XM-30 (PPK-Laser, Kazan, Russia) with the following parameters of laser processing (LP): $\lambda$—wavelength 1.064 $\mu$m, $\upsilon$—radiation frequency (20 kHz), d—laser beam diameter (0.01 mm), l—distance between linear trajectories (0.01 mm), varying $\nu$—the speed of movement of the laser beam (300 $\div$ 700 mm/s) and W—laser power (6 $\div$ 12 W). LP was performed with a single laser pass, i.e., with obtaining a linear "bed" structure. Furthermore, to remove the metal dust formed during the LP process, the samples were washed with ethanol and dried in air at $t = 65$ °C.

Ethanol solutions of octadecylphosphonic (ODPA, 97%, SigmaAldrich, Darmstadt, Germany) and stearic (SA, 98.5%, SigmaAldrich, Darmstadt, Germany) acids and vinyltrimethoxysilane (VS, 98%, Penta-91, Moscow, Russia) were used as hydrophobic agents. The surface modification of 6063 alloy samples was carried out under static conditions at $t = 20 \pm 2$ °C for 60 min, then the samples were dried at $t = 65$ °C for 60 min and washed with distilled water to remove physically adsorbed layers.

### 2.3. Estimation of Surface Roughness Parameters for Alloy 6063 Specimens

#### 2.3.1. Optical Microscopy

Micrographs of the surface of alloy 6063 after mechanical polishing, etching, and LP were obtained using an optical microscope Biomed-3 (Biomed, Saint Petersburg, Russia) equipped with a camera at $\times 100$ magnification.

#### 2.3.2. Profilometry

To assess the parameters of the surface roughness, a Model-130 profilometer (PROTON, Zelenograd, Russia) was used and the index was calculated according to the system of middle lines. The principle of operation of the profilometer consists in sequential tracing of the sample with a diamond needle located perpendicular to the surface under study, and converting its vibrations into a digital signal with subsequent signal processing on a computer. The length of the measurement path was 12.5 mm; at least five measurements of the surface profile were carried out for each sample. The roughness class was determined according to ISO 1302: 2002 and the average values of the parameters were calculated: $R_z$, $\mu$m—the height of the profile roughness; $R_a$, $\mu$m—arithmetic mean deviation; and the roughness class. Roughness parameters were determined for samples subjected to mechanical polishing, alkaline etching and LP.

### 2.4. Characterization of SHP Surface Alloy 6063

2.4.1. Water Contact Angle Measurements and Their Temporary Evolution

To measure $\theta_c$ by the static method, the samples were placed in a laboratory setup with a Levenhuk M 1000 Plus camera (Levenhuk, Tampa, FL, USA) and a drop of distilled water (3–5 μL) was placed on the metal surface. Determination of $\theta_c$ was carried out from photographic images of a drop using a graphical editor. To obtain reliable wetting characteristics, three samples of each type of coating were prepared and the initial $\theta_c$ was measured 5–10 s after the droplet landing on five different areas of the surface of each sample. The average value of the angle was determined for 10 successive images of the drop. The absolute measurement error $\theta_c$ was $\pm 2°$.

In the study of the kinetics of degradation of the obtained SHP layers in time upon exposure of the samples to distilled water, the change in $\theta_c$ was estimated. When testing samples that lasted up to 70 days, $\theta_c$ measurements were performed every 7 days. The samples were removed from distilled water, dried at $t = 65\ °C$, $\theta_c$ was determined and then the samples were returned to the solution to continue testing.

2.4.2. XPS Studies

An Omicron ESCA+ spectrometer (Omicron NanoTechnology, Taunusstein, Germany) was used to analyze the composition of surface layers. The pressure in the analyzer chamber did not exceed $8 \cdot 10^{-10}$ mbar. The source of radiation was a Mg-anode (MgK$\alpha$ 1253.6 eV, power of 252 W, Omicron NanoTechnology, Taunusstein, country). The pass energy was 10 eV. A flood gun was used to compensate for the charge on samples. The spectra positions were standardized by the C1s peak position of hydrocarbon contaminations, which was accepted 285.0 eV.

The spectra were processed by the Unifit 2009 program (Unifit-software, Leipzig, Germany). Fitting of spectra was carried out after the subtraction of the background determined by Shirley's method [47]. The peak position was determined with an accuracy of 0.1 eV. The atomic ratios of elements were calculate using the integral intensities under the peaks, taking into account the photo-ionization cross-sections $\sigma$ of the corresponding electronic shells [48].

The samples for the X-ray photoelectron spectroscopy studies were made of aluminum alloy 6063 plates of $10 \times 20$ mm$^2$, which, after obtaining SHP layers of ODPA on them, were washed with distilled water in an ultrasonic bath for 2 min and then transferred to the spectrometer chamber. The regions of C1s (302–280 eV), O1s (540–525 eV), P2p (144–124 eV) and Al2p-electrons (80–68 eV) were analyzed to qualitatively study the films.

### 2.5. Study of the Protective Properties of SHP Films Formed on Alloy 6063

2.5.1. Polarization and EIS Measurements

The test solution for electrochemical measurements was a borate buffer solution with pH 7.4 (0.2 M $H_3BO_3$ + 0.05 M $Na_2B_2O_7 \times 10\ H_2O$; 99.5%; RusKhim, Moscow, Russia), containing 10 mmol·L$^{-1}$ NaCl (chemically pure, RusKhim, Moscow, Russia). Electrode potentials ($E$) were measured relative to a saturated silver chloride electrode and recalculated to the normal hydrogen scale. The counter electrode was the platinum wire. The electrode working surface area was 1 cm$^2$.

The polarization curves of the 6063 alloy were recorded in a clamping cell using an Autolab PGSTAT302 potentiostat (Metrohm, Utrecht, The Netherlands). Samples without and with preformed SHP films were immersed in a test solution and kept for 60 min until $E_{cor}$ was established, and then cathodic/anodic polarization was performed. The protective properties of the SHP layers were assessed by the increase in the value of the local depassivation potential $E_{pt}$ in comparison with a value measured on the electrode without SHP treatment.

Electrochemical impedance spectroscopy (EIS) was performed using an Autolab PG-STAT302 potentiostat equipped with an FRA32M module (Metrohm, Utrecht, the Netherlands). The impedance spectra were recorded in the frequency range $f$ from 10 kHz to 0.1 Hz

with AC voltage amplitude of 10 mV. The results were processed using the NOVA software. The experimental data were in agreement with the calculated data, by at least 95%.

### 2.5.2. Corrosion Test in a Salt Spray Chamber

The protective ability of the formed SHP coatings was determined by testing the samples in a Weiss SC/KWT 450 salt spray chamber (SSC) (Weiss Technik GmbH, Reiskirchen, Germany) according to ISO 9227. A 5% NaCl solution (pH 6.9) was used as a saline solution in SSC. The chamber operated continuously in a cyclic mode (one cycle—15 min of spraying the saline solution, then the chamber was turned off for 45 min, after which the cycle was repeated). The tests were carried out at $t$ = 35 °C and relative air humidity H = 95–100. The samples were examined three times a day to establish the time until the first signs of corrosion appeared—$\tau_{cor}$.

At the same time, $\theta_c$ was measured every 7 days on control samples of the alloy with SHP coatings during tests in SSC, the samples were washed with distilled water, dried at $t$ = 65 °C, $\theta_c$ was determined and then the samples were returned to the SSC to continue testing.

## 3. Results and Discussion

### 3.1. Surface Morphology of Alloy 6063

Figure 1 shows micrographs of the surface of alloy 6063 after mechanical polishing (a), etching in a 10% NaOH solution (b) and LP (c). On the grinded surface of the alloy, there are small surface defects and traces of mechanical cleaning. As a result of alkaline etching, the surface heterogeneity of the 6063 alloy increases, and a structure with a complex relief is formed. After the LP of the alloy with the following parameters: W = 9 W and ν = 300 mm/s; ordered periodic structures are formed on the surface.

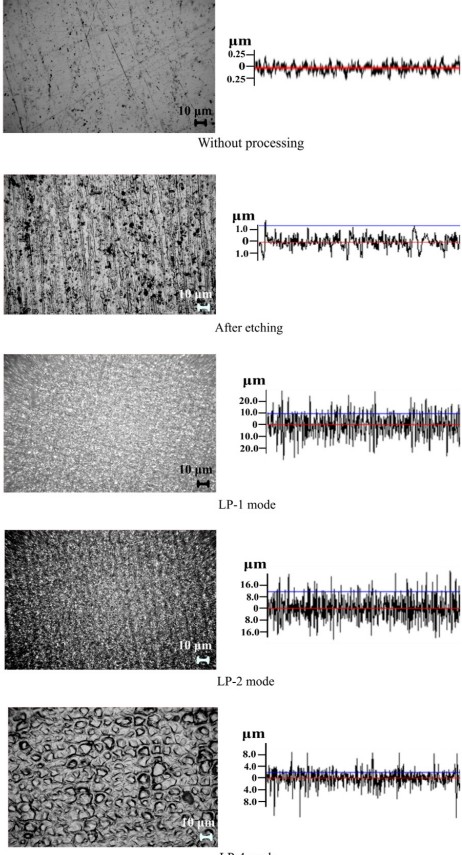

**Figure 1.** Micrographs (magnification ×100) and profilograms of the surface of alloy 6063.

According to the data of profilometric measurements (Table 1), the height of the profile roughness of mechanically polished samples does not exceed 0.46 µm, with an arithmetic mean deviation of 0.04 µm, which corresponds to the second class of roughness. With alkaline etching, the surface inhomogeneity increases: $R_z$ = 1.95 µm, $R_a$ = 0.18 µm, roughness class 4.

**Table 1.** Results of profilometric measurements of the surface of alloy 6063 samples after mechanical polishing, etching in 10% NaOH solution and subjected to LP. Measurement error of profilometer is 2%.

| Treatment | | | Measurement Parameters | | |
|---|---|---|---|---|---|
| | | | $R_z$, µm | $R_a$, µm | Roughness Class |
| After mechanical polishing | | | 0.46 | 0.04 | 2 |
| After etching in 10% NaOH | | | 1.95 | 0.18 | 4 |
| LP mode | Speed, mm/s | Power, W | | — | |
| LP-1 | 300 | 9 | 14.70 | 2.52 | 9 |
| LP-2 | 500 | 9 | 9.82 | 1.82 | 8 |
| LP-3 | 700 | 9 | 9.25 | 1.72 | 8 |
| LP-4 | 500 | 6 | 4.05 | 0.43 | 6 |

With the LP of the surface of alloy 6063, the surface inhomogeneity increases significantly in comparison with mechanically polished and etched samples. Thus, for the LP-1 mode (W = 9 W, ν = 300 mm/s), the value $R_z$ = 14.70 µm (9th class of roughness). An increase in the scan rate of the laser beam to 500 and 700 mm/s (modes LP-2 and LP-3) leads to a decrease in the value of the parameter $R_z$ to 9.82 and 9.25 µm, respectively. LP-4 mode (W = 6 W, ν = 500 mm/s) gives lower roughness: the average roughness size does not exceed 4.05 µm (roughness class 6).

*3.2. SHP State of the Coatings and Its Evolution of during Long-Term Contact with Water Solution*

The surface of alloy 6063 after mechanical polishing is hydrophilic, and the contact angle is 45° (Table 2). The formation of a surface with microstructural roughness on the alloy leads to an increase in hydrophilic properties. After a short-term etching of the alloy in a 10% NaOH solution, the surface is characterized by a low value of $\theta_c$ = 35°. After LP, the alloy surface becomes superhydrophilic, regardless of the LP mode.

**Table 2.** Change in the contact angle $\theta_c$ on alloy 6063 from preliminary surface preparation and then after its modification with an ethanol solution of ODPA $C$ = 1 mmol·L$^{-1}$.

| Surface Preparation | | Treatment in ODPA |
|---|---|---|
| — | $\theta_c$, degree | $\theta_c$, degree |
| Mechanical polishing | 45 ± 2 |  122 ± 2 |
| Etching in 10% NaOH | 35 ± 2 |  162 ± 2 |

**Table 2.** *Cont.*

| Surface Preparation | | Treatment in ODPA |
|---|---|---|
| − | $\theta_c$, degree | $\theta_c$, degree |
| LP-1 | ≤2 |  162 ± 2 |
| LP-2 | ≤2 |  168 ± 2 |
| LP-3 | ≤2 |  165 ± 2 |
| LP-4 | ≤2 |  161 ± 2 |

Subsequent the exposure of alloy samples, preliminarily subjected to etching or LP, in an ethanol solution of 1 mM ODPA, leads to their superhydrophobization (Table 2). It should be noted that the most effective pretreatment is LP-2: $\theta_c$ = 168°, which is obviously associated with a more uniform surface morphology (Figure 1).

Of great importance for the SHP coatings is the keeping of their SHP properties for a long time under the influence of corrosive factors. The coatings obtained on the etched surface of alloy 6063 have the lowest stability in an aqueous solution (for 7 days). For samples subjected to LP and then modified in a solution of 1 mM ODPA SHP, the state persists for a long time, and for the best mode (LO-2), SHP is retained for 32 days (Figure 2a). Further exposure of the samples in $H_2O$, although accompanied by a decrease in $\theta_c$, but even after 70 days of testing, the surface retains its hydrophobic properties. For samples subjected to LP-2 and then treated in ODPA solution, $\theta_c$ = 128°.

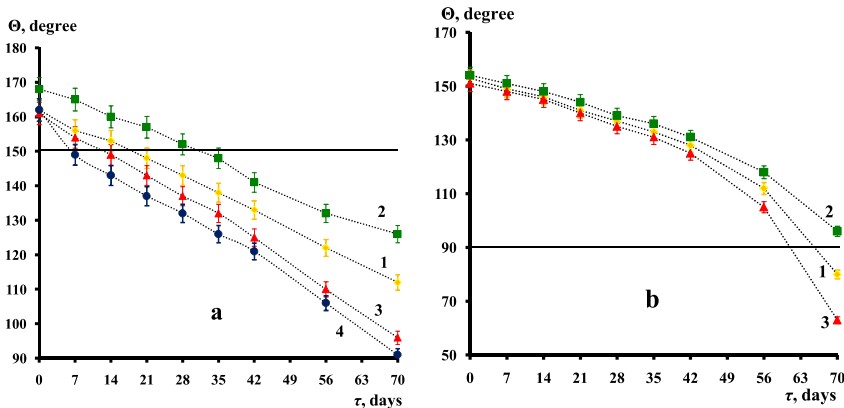

**Figure 2.** Change in the contact angle $\Theta_c$ on samples of alloy 6063 versus the time of holding the samples in distilled water, preliminarily subjected to LP (**1**—LP-1; **2**—LP-2; **3**—LP-4) or etched in a 10% solution of NaOH (**4**), and then modified in an ethanol solution containing 1 mmol·L$^{-1}$ ODPA (**a**) and 10 mmol·L$^{-1}$ SA (**b**).

Exposure of laser-textured samples of alloy 6063 in ethanol solution containing 10 mmol·L$^{-1}$ of SA also leads to its SHP, regardless of the used LP regimes, and $\theta_c$ = 151° ÷ 154° (Figure 2b).

After immersion of alloy 6063 specimens, preliminarily subjected to LP, and then treated in an ethanol solution of 10 mM SA, the SHP state remains only for 7 days of testing (Figure 3). With an increase in the duration of the tests, the value of $\theta_c$ continues to decrease, but by no more than 3°–5° from each subsequent measurement in the first 42 days of testing. Furthermore, the destruction of the SA coating in an aqueous solution occurs more intensively and after 70 days of testing only for the LP-2 mode the SA coating retains its hydrophobic properties, and $\theta_c$ = 96°.

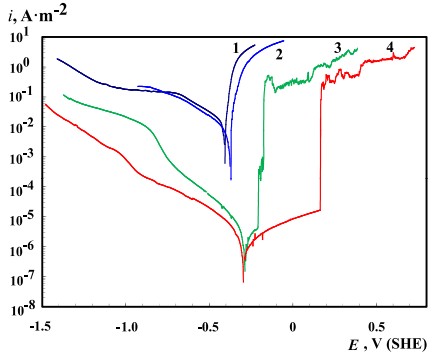

**Figure 3.** Polarization curves of 6063 alloy in a borate buffer solution (pH 7.4) containing 1 mmol·L$^{-1}$ NaCl, after LP-2 and SHP treatment in the solutions of CI (*C*, mmol·L$^{-1}$): **1**—without treatment; **2**—after LP-2; **3**—10 SA; **4**—1 ODPA.

According to [49] the hydrophobic properties of the compound can be estimated using the logarithm of its distribution coefficient in the system of two immiscible liquids (octanol-water)—lg *P*. The calculated values showed that SA (log *P* = 8.23) is more hydrophobic than ODPA (lg *P* = 7.01).

Despite the lower hydrophobicity of ODPA molecules, according to [50], it forms a hydrophobic self-organized monolayer on the oxidized aluminum surface, which is better ordered than the layer of higher carboxylic acid. This can be explained by the stronger interaction of the phosphonic group with the surface, in comparison with the carboxylic group, due to the higher polarity, which can have a significant effect on the degradation rate of SHP coatings in an aqueous solution.

It is significant that signs of corrosion on all samples with SA and ODPA coatings do not appear in water during the entire test period. On untreated samples, corrosion was observed in the form of surface tarnishing after 24 h as a result of surface oxidation.

It can be concluded that the LP-2 mode is the most effective. Probably, the LP-2 treatment mode leads to a formation of a surface with a more uniform morphology, which provides better conditions for the formation of SHP layers after the adsorption of anionic compounds with a hydrophobic alkyl chain.

### 3.3. Polarization Measurements

Polarization measurements show (Figure 3) that LP has an insignificant effect on the electrochemical behavior of the alloy in comparison with a mechanically cleaned sample. The subsequent formation of a SA or ODPA layer on the laser-modified surface provides a significant decrease in the current densities. In this case, the form of the anodic curve after the formation of a hydrophobizing film is characterized by a certain value of the potential, at which a sharp increase in the anodic current occurs—the breakdown potential ($E_{pt}$). The results confirm the high strength of the ODPA layer, for which $E_{pt} = 0.15$ V, exceeding the value of $E_{pt}$ for SA by 0.17 V.

### 3.4. EIS

The EIS results confirm the results of polarization measurements, and also demonstrate a certain difference in the behavior of the studied samples (Figure 4). The calculated parameters of the equivalent electrical circuits used to describe the spectra are presented in Table 3. The circuit contains the following elements: $R_{sol}$—solution resistance; $Q_f$—constant phase element (CPE), describing the capacitance of the surface film; $R_f$—film resistance; $Q_{ox}$—constant phase element (CPE) describing the capacitance of the oxide layer; $R_{ox}$—resistance of the oxide layer; $Q_{dl}$—CPE parameter, describing the capacitance of the electric double layer; $R_p$—polarization resistance; W—Warburg impedance simulating the diffusion process. It follows from the obtained data that the ODPA film, in comparison with the SA, exhibits large impedance values, exceeding it in $R_p$ by almost an order of magnitude. The $Q_f$ and $Q_{dl}$ values are also lower in the case of ODPA, which indicates a decrease in the electrochemical activity of the surface. Additionally, in the case of an ODPA film, the role of diffusion processes increases, which indirectly indicates its higher barrier properties.

**Table 3.** Estimated parameters of equivalent electrical circuits used to describe the results of the EIS.

| − | − | LT | LT-SA | LT-ODPA |
|---|---|---|---|---|
| $R_s$, $\Omega$ cm$^2$ | − | 745.54 | 963.81 | 595.32 |
| $Q_f$ — Y, ($S\ s^n$ cm$^{-2}$) | | − | $7.0009 \times 10^{-9}$ | $5.4734 \times 10^{-10}$ |
| $Q_f$ — n | | - | 0.90521 | 0,97652 |
| $R_f$, $\Omega$cm$^2$ | − | - | $3.0145 \times 10^5$ | $8.734 \times 10^6$ |
| $Q_{ox}$ — Y($S\ s^n$ cm$^{-2}$) | | - | $5.0584 \times 10^{-8}$ | - |
| $Q_{ox}$ — n | | - | 0,66644 | - |
| $R_{ox}$, $\Omega$ cm$^2$ | − | - | $5.9351 \times 10^6$ | - |
| $Q_{dl}$ — Y ($S\ s^n$ cm$^{-2}$) | | $2.7871 \times 10^{-5}$ | $3.2058 \times 10^{-7}$ | $5.1606 \times 10^{-9}$ |
| $Q_{dl}$ — n | | 0.89962 | 0.74578 | 0.62758 |
| $R_p$, $\Omega$ cm$^2$ | − | $1.3089 \times 10^5$ | $6.167 \times 10^7$ | $4.0889 \times 10^8$ |
| W, ($S\ s^{1/2}$) | − | - | - | $3.1653 \times 10^{-8}$ |



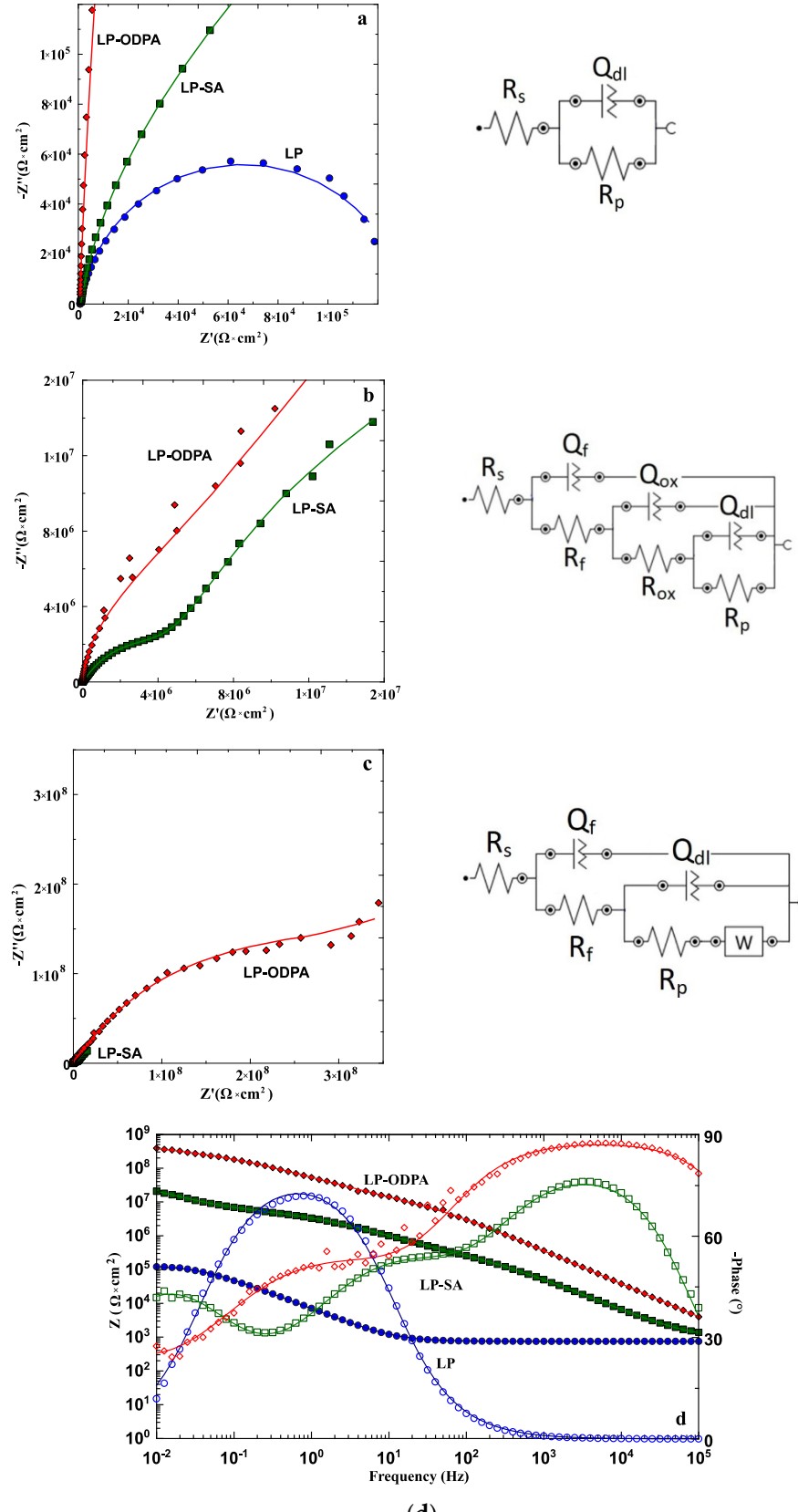

**Figure 4.** Nyquist (**a–c**) and Bode (**d**) diagrams obtained in a borate buffer solution on alloy 6063 after LP, as well as after LP and subsequent treatment with SA and ODPA.

*3.5. XPS*

To determine the peak positions and their succeeding displacement, XPS measurements were carried out on reference samples.

According to XP-spectra of powder of ODPA, the peak positions of main elements were as follows: for hydrocarbon radical at the binding energy of 285.0 eV [49]; for the C-P bond 286.1 eV; the difficult O1s peak was split into two peaks, one of which has a maximum at 533.3 eV and the other was at 531.8 eV [50] for P-OH and P = O bonds, respectively; the P2p doublet was in the position of the maximum of the $P2p_{3/2}$ peak at 134.2 eV.

The adsorption of ODPA on the aluminum alloy 6063 surface was indicated by the presence of asymmetric peak in the P2p phosphorus region and big peak of hydrocarbon radical. This asymmetry is associated with the spin—orbit splitting of the 2p electronic level and a small shift between the $2p_{3/2}$ and $2p_{1/2}$ components of the doublet, which is 0.8–1 eV [51,52]. In this case, the peak position of the $P2p_{3/2}$ electrons was at 133.7 eV and P2p1/2 was at 134.6 eV. This $P2p_{3/2}$ maximum was occurred at a lower binding energy than the phosphorus peak of the ODPA powder. This shift can be explained by the formation of a chemisorbed layer of ODP-Al.

Figure 5 shows the fitting Al 2p spectra obtained for the bare sample (a) and after the ODPA layer formation (b). On the bare sample, the fitting shows the presence of an alumina $Al^{3+}$ peak at 74.4 eV and an Al metal peak at 71.6 eV [53,54]. These peak positions are in good agreement with the data on the shift between these states, which was 2,8 eV [55]. In the Al2p spectrum of the sample, after the formation of the ODPA layer, a shift of the spectrum maximum to the region of higher binding energies by 0.4 eV is observed. In this spectrum, in addition to two primary states, a third is observed at 75.2 eV. This peak can be attributed to the ODP-Al bond. However, after the adsorption of this substance, the full width at half maximum (FWHM) of $Al^{3+}$ decreased. The values of FWHM were presented in Table 4. This can be explained by a decrease in the proportion of aluminum hydroxide, which is also confirmed by the oxygen spectra.

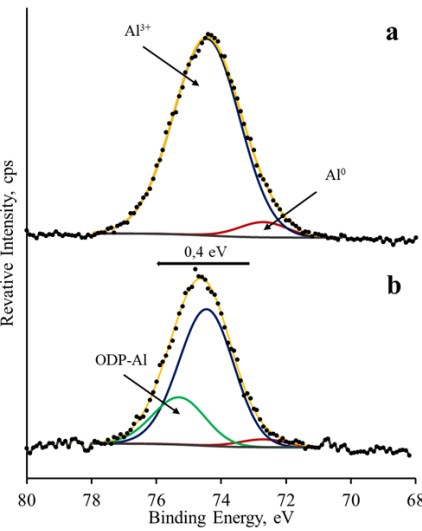

**Figure 5.** Al2p core level XPS spectra of the bare sample (**a**) and after the ODPA layer formation (**b**).

**Table 4.** Values of FWHM for the bare sample (1) and after the ODPA layer formation (2).

| States | $Al^0$ | $Al^{3+}$ | ODP-Al |
|:------:|:------:|:---------:|:------:|
| 1 | 1.75 | 2.44 | — |
| 2 | 1.75 | 2.09 | 2.03 |

Figure 6 shows the O1s spectra of study samples. On the bare sample peak (a), three states were observed, i.e., $Al_2O_3$ at 530.2 eV, $Al(OH)_3$ at 531.7 eV and adsorbed water

vapors at 533.3 eV [56]. In contrast to the bare sample, on the after ODPA adsorption sample, there were two additional peaks. The maxima of these peaks are at 532.2 eV and 530.9 eV and their ratio is 2:1. This is similar to the peaks of oxygen in powder. The peak at 532.2 eV can be attributed to the P-OAl bonds and the peak at 530.9 eV can be attributed to the P=O bond. The FWHM values of the initial states were 2.4 and 2.6 eV for oxide and hydroxide, respectively. For the P-OAl and P=O bonds, these values were 2 eV. The atomic concentrations of the oxide and hydroxide states are presented in Table 5. According to these data, the Al(OH)$_3$ part was also decreased, which indicates the formation of a bond of ODPA molecules with the surface hydroxide layer.

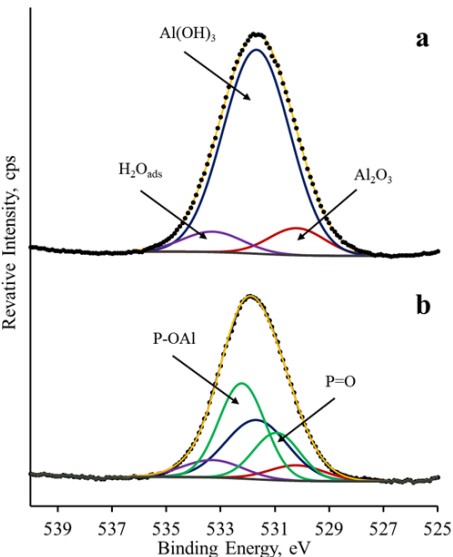

**Figure 6.** O1s core level XPS spectra of the bare sample (**a**) and after the ODPA layer formation (**b**).

**Table 5.** The atomic concentrations (%) of the oxide and hydroxide states of the bare sample (1) and after the ODPA layer formation (2).

| States | Al$_2$O$_3$ | Al(OH)$_3$ |
| --- | --- | --- |
| 1 | 9.98 | 90.02 |
| 2 | 18.72 | 81.28 |

*3.6. Corrosion Tests and a Degradation of SHP Coatings*

The salt spray corrosion tests confirm the high protective ability of SHP films. On the surface of the samples pretreated with LP-1, the first corrosion damages were found after 1.5 h of testing. Subsequent surface modification in an ethanol solution of 10 mM SA increases $\tau_{cor}$ up to 15 days (Figure 7a). We have previously shown [46] that layer-by-layer treatment of the surface of this alloy (first in an ethanol solution of 10 mM VS, and then in 10 mM SA) increases the stability of the SHP layers. It also increases the protective ability of SHP films on the surface previously exposed to LP-1, which can be seen from the results of tests in SSC: $\tau_{cor}$ = 19 days. However, the best protective ability is possessed by films formed from ODPA solutions: $\tau_{cor}$ = 23 days. For comparison, on the sample after mechanical cleaning, the first corrosion damage was detected after 8 h, i.e., LP itself worsens the corrosion resistance of the alloy due to the formation of a more developed and hydrophilic surface. After treatment of the cleaned 6063 in SA solution, $\tau_{cor}$ = 18 h (0.75 days), and for ODPA, $\tau_{cor}$ = 26 h (1.08 days). Thus, the preliminary LP of the sample surface and the imparting of the properties of the SHP to it makes it possible to increase the corrosion resistance of the alloy (the protective ability of the SA and ODPA films) by more than 20 times.

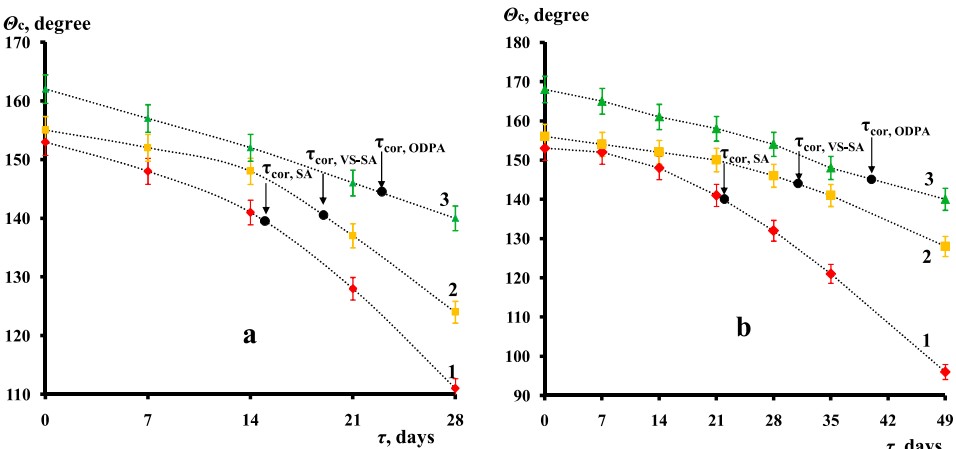

**Figure 7.** Change in the contact angle $\theta_c$ on alloy 6063 samples from the time of exposure of the samples in SSC, previously subjected to LP-1 (**a**) and LP-2 (**b**) and then modified in ethanol solutions containing: **1**—10 mM SA; **2**—layer by layer 10 mM vs. and 10 mM SA; **3**—1 mM ODPA. The arrow in the diagrams is the time of appearance of the first corrosion damage on the samples.

The preliminary LP-2 and subsequent SHP in solutions of SA and ODPA are more effective not only according to the kinetics of degradation of coatings in an aqueous solution, but also under more severe conditions of SSC. On samples subjected to LP-2, the corrosion damage appears in 2.3 days. Subsequent surface treatment in an ethanol solution of 10 mM SA increases $\tau_{cor}$ to 22 days (Figure 7b), and with layer-by-layer modification of 10 mM vs. and 10 mM SA, $\tau_{cor}$ increases to 31 days. Formation of ODPA film on LP-2 treated samples provides maximum effect with $\tau_{cor} = 40$ days.

The results of measuring $\theta_c$ in time on samples 6063 with formed coatings during their tests in SSC indicate the important role of their SHP properties. For samples subjected to LP-1 and modified with 10 mM SA, the SHP state is retained only for 5–7 days of testing, but with layer-by-layer treatment with 10 mM vs. and 10 mM SA, SHP remains for 14 days (Figure 8a). For samples modified with 1.0 mM ODPA, SHP is retained for 19 days.

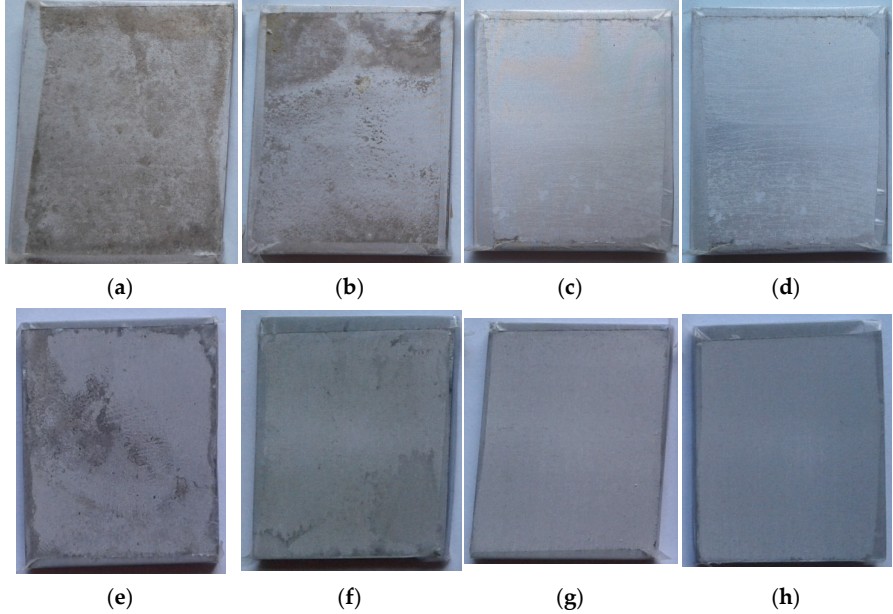

**Figure 8.** Images of 6060 alloy samples preliminarily subjected to LP-1 (**a**–**d**) and LP-2 (**e**–**h**) and then modified by: 10 SA (**b**,**f**); layer by layer 10 vs. and 10 SA (**c**,**g**); ODPA (**d**,**h**) after 35 days of testing in a salt spray chamber.

For LP-2 mode, the change in SHP properties occurs over a longer time (Figure 7b). For samples modified with SA, the loss of SHP properties is observed after 10 days, and with a layer-by-layer modification of vs. and SA after 22 days. For the coating formed in 1.0 mM ODPA, SHP is retained for 35 days.

From Figure 7, it is easy to see that the appearance of the first corrosion damage is observed when $\theta_c$ falls in the range 140°–150°, which is local damage (pitting) (Figure 8). For example, for specimens of alloy 6063 subjected to LP-2 and then modified with 1.0 mM ODPA, the growth of corrosion damages over the entire surface of the specimen is observed only after 65 tests.

Obviously, in all the cases, the protective properties of the films correlate with their hydrophobicity: the higher the value of $\theta_c$ and the slower its change in time, the longer the protection time. All results confirm the greater stability and protective properties of the layers formed by ODPA on the laser-modified surface.

## 4. Conclusions

Alkaline etching and laser texturing of the surface of alloy 6063, and its subsequent modification in solutions of SA and ODPA, lead to its SHP. According to the results of the kinetics of degradation of the SHP state in an aqueous solution, the most optimal mode of laser texturing is LP-2. Under the conditions of water exposure, a gradual degradation of SHP coatings occurs, but even after 70 days of testing, the films formed from ODPA solutions remain hydrophobic and are significantly more stable than the coatings obtained by treating samples with SA solutions.

Electrochemical tests of the 6063 alloy in a chloride-containing borate buffer showed that the SHP coatings formed by SA and ODPA effectively prevent its local depassivation by chlorides. The EIS results confirm the results of polarization and corrosion tests, showing a significant increase in the corrosion resistance of the alloy after adsorption of SA and ODPA on its surface, with a great advantage of ODPA.

The ODPA chemisorption is indicated by the presence of a phosphorus peak and a large hydrocarbon peak on alloy surface spectra after sample wash. The shift of $P2p3_{/2}$ peak maximum indicates an interaction between ODPA and the surface layer. According to the displacement of P-OH in powder and on the alloy surface the bond is formed due to the deprotonation of the OH groups of the inhibitor molecules and their interaction with aluminum hydroxide.

The results of corrosion tests under the harsh conditions of SSC indicate the high protective properties of the SHP coatings. For the coating formed by ODPA on the surface with LP-2, the time to the appearance of the first corrosion damage exceeds 40 days, and their growth is observed after 65 days of testing. It is essential that the SHP state of the formed coating is retained for 35 days.

Despite certain disadvantages of surface texturing, such as increased costs for the process and the difficulty of uniform processing of complex-shaped products, it can be used to significantly improve the anti-corrosion properties of thin films of hydrophobizing agents on metals due not only to the barrier properties of the film, but also to limiting the contact of the metal surface with the corrosive environment. Further improvement of this method may be associated with a more detailed study of the properties and characteristics of a textured surface and the choice of optimal parameters for its preparation, the search for synergistic additives to hydrophobizing agents and layer-by-layer modification of the metal surface, as was shown in the case of SA and vinyltrimethoxysilane.

**Author Contributions:** Conceptualization, A.M.S. and Y.I.K.; methodology, A.M.S., A.A.C. and O.Y.G.; validation, A.M.S., A.A.C. and O.Y.G.; investigation, A.M.S., A.A.C. and O.Y.G.; writing—original draft preparation, A.A.C. and O.Y.G.; writing—review and editing, A.M.S. and Y.I.K.; visualization, A.M.S., A.A.C. and O.Y.G.; supervision, Y.I.K.; project administration, A.A.C. All authors have read and agreed to the published version of the manuscript.

**Funding:** This research was funded by the Ministry of Science and Higher Education of the Russian Federation ("Chemical resistance of materials, protection of metals and other materials from corrosion and oxidation" 122011300078-1).

**Institutional Review Board Statement:** Not applicable.

**Informed Consent Statement:** Not applicable.

**Data Availability Statement:** Not applicable.

**Conflicts of Interest:** The authors declare no conflict of interest.

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
