# Peer review of "Stability of Supehydrophobic Layers Formed by Organic Acids on the Surface of Aluminum Alloy 6063"

_coatings, doi:10.3390/coatings12101468_

Round 1

Reviewer 1 Report

The manuscript describes des stability superhydrophobic layers formed by organic acids on the surface of a selected aluminium alloy.

With regard to the experiments design there are some concerns. For instance it is described in the methods that at least five measurements of the surface profile were carried out for each sample or the contact angle was measured in 5 different areas of the surface of each sample. However, it appears measurements were not repeated on independent samples. I would ask the author either to correct the text or repeat the experiments to she the reproducibility for their results. Some of the results are also not including the deviations from the obtained value, for example Table 1.

Additionally, the  Figures of the manuscript can be improved in terms of the readability, different scales, scale bars and the captions should be extended so they become self-explanatory. A little overview about the different modes would also help the reader   

Author Response

The authors would like to thank the reviewer for careful consideration of the article and important remarks. We have made a number of corrections to the text and figures, and would also like to provide clarifications on the comments.

The influence of processing modes on surface characteristics was not considered in detail within the framework of this work, with the exception of roughness parameters. We were more interested in the stability of the anticorrosive and superhydrophobic properties of films obtained on the surface of an aluminum alloy with various preliminary preparations. In further work on improving the methods for obtaining such films, the influence of the metal surface morphology will be studied in more detail.

Reviewer 2 Report

The authors have discussed the superhydrophobic treatment of Aluminum 6063. The presentation and writing are good. However, there are many issues need to be addressed before publication. I recommend a major revision for this article.

1.       The introduction seems very lengthy, and the problem statement is unclear. I am afraid that this article may attract a specific group of audience working on the superhydrophobic treatment of metals.

2.       What is the importance of selecting Aluminum 6063 for this study? This needs to be clearly discussed in the introduction part.

3.       It is well known that surface roughness plays an important role in determining its superhydrophobicity. The authors are requested to provide SEM and AFM images of the coated and uncoated surfaces to justify the roughness. However, I think the presence of dual (nano and micro) scale roughness is required to attain the water contact angle of >150°. The most exemplary case is the lotus leaf.

4.       This comment is in relation to the previous one. Below are some relevant articles; the authors may go through them carefully and incorporate them accordingly.

a.       https://doi.org/10.1021/acsbiomaterials.8b00209

b.       https://doi.org/10.1080/03602559.2018.1447128

c.       https://doi.org/10.1016/j.apsusc.2016.04.101

d.       https://doi.org/10.1002/slct.202001092

e.       https://doi.org/10.1007/s11831-021-09689-1

f.        https://doi.org/10.1021/acs.iecr.7b00225

g.       https://doi.org/10.1016/j.porgcoat.2022.107062

5.       In Table 2 first row, the Ɵс, degree value is written as 45÷50. Kindly correct the mistake.

6.       Some figures can be clubbed together to reduce the total number of figures, such as Figures 2 and 3, Figures 6 and 7, etc.

7.        The work must have some limitations and future work. It would be better if the authors could discuss them in the conclusion part.

8.       Minor grammatical check and corrections.

Author Response

The authors would like to thank the reviewer for careful consideration of the article and important remarks. We have made a number of corrections to the text and figures, and would also like to provide clarifications on the comments.

  1. The aim of the work was not so much the phenomenon of superhydrophobicity, but its stability when protecting an aluminum alloy from the effects of corrosive factors, and the article primarily has a “corrosion” direction.
  2. Alloy 6063 is one of the widely used aluminum alloys and it finds application in architectural and building products, in electrical components and conduits, in pipes and tubes, in door and window. The introduction part was supplemented.
  3. First of all, we were interested in the possibility of obtaining superhydrophobic layerswith high anticorrosion properties and stability in order to understand the prospects of such a protection method. The effect of processing modes and surface conditions on the characteristics of hydrophobic films is the subject of further research.

      4-6. Some changes were made.

  1. The conclusion part was supplemented.

Round 2

Reviewer 1 Report

Following the reviewers suggestions, the manuscript has improved.